# Outcomes of Herpes Simplex Virus Pneumonitis in Critically Ill Patients

**DOI:** 10.3390/v14020205

**Published:** 2022-01-21

**Authors:** Wen-Jui Chang, Hsin-Yao Wang, Yu-Chen Huang, Chun-Yu Lin, Shaw-Woei Leu, Meng-Jer Hsieh, Chung-Chi Huang

**Affiliations:** 1Department of Thoracic Medicine, Chang Gung Memorial Hospital, Taipei City 10507, Taiwan; trevis1118@gmail.com (W.-J.C.); yuchenhahaha@gmail.com (Y.-C.H.); pitiful1984@gmail.com (C.-Y.L.); swleu@cgmh.org.tw (S.-W.L.); mengjer@yahoo.com (M.-J.H.); 2College of Medicine, Chang Gung University, Taoyuan 33305, Taiwan; 3Department of Laboratory Medicine, Chang Gung Memorial Hospital at Linkou, Taoyuan 33305, Taiwan; mdhsinyaowang@gmail.com; 4Ph.D. Program in Biomedical Engineering, Chang Gung University, Taoyuan 33305, Taiwan; 5Division of Pulmonary and Critical Care Medicine, Chang Gung Memorial Hospital, 5, Fu-Xing Street, Kweishan, Taoyuan 33305, Taiwan

**Keywords:** herpes simplex virus (HSV) pneumonitis, acute respiratory distress syndrome (ARDS), diabetes mellitus (DM), Acute Physiology and Chronic Health Evaluation II (APACHE II) score

## Abstract

Critically ill patients, such as those in intensive care units (ICUs), can develop herpes simplex virus (HSV) pneumonitis. Given the high prevalence of acute respiratory distress syndrome (ARDS) and multiple pre-existing conditions among ICU patients with HSV pneumonitis, factors predicting mortality in this patient population require further investigation. In this retrospective study, the bronchoalveolar lavage or sputum samples of ICU patients were cultured or subjected to a polymerase chain reaction for HSV detection. Univariable and multivariable Cox regressions were conducted for mortality outcomes. The length of hospital stay was plotted against mortality on Kaplan–Meier curves. Among the 119 patients with HSV pneumonitis (age: 65.8 ± 14.9 years), the mortality rate was 61.34% (73 deaths). The mortality rate was significantly lower among patients with diabetes mellitus (odds ratio [OR] 0.12, 95% confidence interval [CI]: 0.02–0.49, *p* = 0.0009) and significantly higher among patients with ARDS (OR: 4.18, 95% CI: 1.05–17.97, *p* < 0.0001) or high (≥30) Acute Physiology and Chronic Health Evaluation II scores (OR: 1.08, 95% CI: 1.00–1.18, *p* = 0.02). Not having diabetes mellitus (DM), developing ARDS, and having a high Acute Physiology and Chronic Health Evaluation II (APACHE II) score were independent predictors of mortality among ICU patients with HSV pneumonitis.

## 1. Introduction

The herpes simplex virus (HSV) is associated with various infections, including respiratory infections [1,2], causing fever, sore throat, gingivostomatitis, and localized lymphadenopathy [2,3,4]. Tuxen et al. first defined HSV pneumonitis as occurring in immunocompromised patients, requiring evidence of pulmonary parenchymal invasion, and contributing to unexplained clinical deterioration [5]. However, HSV can also infect immunocompetent patients with critical illness [6,7]. HSV isolated from lung secretions serves as a marker of severe illness, even in the absence of a firm diagnosis of HSV-related pneumonia or of evidence of HSV pneumonitis being a direct cause of death [8,9]. One prospective study showed between 22% and 16% of intensive care unit (ICU) patients had positive test results when cultured from oropharyngeal swabs and bronchoalveolar lavage (BAL), respectively [10]. Another study detected HSV-1 DNA in bronchoalveolar lavage fluid obtained from 32% of the samples recovered from ICU patients, compared to 15% from non-ICU patients [9].

HSV pneumonitis is associated with adult respiratory distress syndrome (ARDS), prolonged ventilator support, and increased mortality in critically ill patients [5,11,12]. One study noted that the incidence of ARDS in patients with HSV pneumonitis was between 30% and 55% [12]. In another investigation, the mortality rates among patients with ARDS who tested negative or positive for HSV were 45% and 60%, respectively [5]. In a study on critically ill patients by van den Brink, a mortality rate of 40% was observed among patients from whom HSV was isolated from bronchoalveolar lavage [8]. The same investigation revealed that patients in the ICU with HSV pneumonitis died of severe pre-existing conditions unrelated to being immunocompromised [8]. Luzzati et al. demonstrated that underlying diseases were related to mortality in patients with HSV latency; however, their study participants included patients in whom only the upper airway was colonized [13]. Few studies have addressed factors associated with mortality in HSV pneumonitis, especially in critically ill patients. Results regarding the effectiveness of antiviral therapy for critically ill patients with HSV pneumonitis have been inconsistent; some studies have reported that acyclovir treatment confers clinical benefits [5,6,14], whereas others have indicated that such treatment is ineffective [8,15,16]. The associations among pre-existing medical conditions, the administration of antiviral agents, and clinical outcomes in patients with HSV in the ICU warrants further exploration. Therefore, we examined the clinical conditions, laboratory test results, and ventilator settings of such patients. We also investigated the factors predicting mortality in this sample.

## 2. Materials and Methods

### 2.1. Study Design and Participants

We reviewed the medical records from Chang Gung Memorial Hospital (CGMH), Taoyuan City, Taiwan; identifying 270 patients in whom HSV had been isolated from the nasal pharynx, throat, or lung between January 2015 and January 2019. The inclusion criteria were an age ≥ 18 years, a BAL or sputum sample testing positive for HSV through polymerase chain reaction (PCR) or culture, and critical condition requiring ICU care. Thirty patients were excluded because they were aged <18 years, 104 patients were excluded because HSV was detected only in samples from the nasal pharynx or throat, and 17 patients were excluded because they were not treated in the ICU during their hospitalization; 119 patients remained (Figure 1).

The patients’ characteristics, namely their age, sex, smoking history, pre-existing conditions, laboratory test results, and history of steroid or immunosuppressant prescription, were recorded. HSV of the lower airways was defined as the positive identification of HSV from BAL or from bronchial aspirate. Five 30 mL BAL samples were obtained from intubated patients using a flexible bronchoscope (BF-P240 or BF-40; Olympus, Tokyo, Japan; outer diameter 6.0–8.0 mm, inner diameter 2.8 mm). The sampling area was determined according to the location of the infiltration on a chest radiograph or the location of a bronchopulmonary segment with visible purulent secretion [17]. The bronchial aspirate or BAL samples were transported to the laboratory in sterile containers, which were cultured or subjected to a PCR for HSV detection. Mucosal hemorrhage was indicated by the observation of nonremovable, noniatrogenic petechiae or hemorrhagic spots in the mucosa of the bronchi and tracheae reachable by the bronchoscope [12]. Orolabial lesions were recorded by a sole blinded investigator and identified as orolabial ulcers [12] or herpetic gingivostomatitis [18] (Figure 2). Disease severity at the time of ICU admission was assessed using the Acute Physiology and Chronic Health Evaluation II (APACHE II) [19]. The ARDS was defined according to the Berlin definition [20]. Acute kidney injury (AKI) was recorded according to the risk, injury, failure, loss, and end stage criteria [21]. The ventilation settings and biochemical findings were obtained at the time of HSV pneumonitis diagnosis. The duration of mechanical ventilation, length of hospital stay, and length of ICU stay were calculated in days. According to the primary outcome, and the mortality during hospital stays, the patients were divided into survivors and non-survivors. The Institutional Review Board (IRB) of CGMH approved the study protocol (IRB number 202001331B0), and the study was performed in compliance with the tenets of the Declaration of Helsinki. Given the retrospective nature of this study and the fact that it had no bearing on modifications to patient management, the IRB waived the need for written informed consent. In addition, all personal information in the database had been encrypted and deidentified.

### 2.2. Statistical Analysis

All data are presented as means ± standard deviations, as medians (interquartile ranges), or as frequencies (percentages). The Mann–Whitney U test and Fisher’s exact test were conducted to determine any significant differences between the continuous and categorical variables, respectively. Univariable and multivariable Cox regression analyses were performed for risk factor evaluation. Risk factors significantly associated with mortality in the univariable regression were included in a stepwise multivariable Cox regression model. We employed the extended Kaplan–Meier method for mortality curve estimation. Receiver operating characteristic (ROC) curves were obtained according to the multivariable Cox regression model, and the area under the curve (AUC) were calculated. Differences were considered significant if *p* < 0.05. The analyses were conducted using R software, version 3.5.2 (R Foundation for Statistical Computing, Vienna, Austria, URL: https://www.R-project.org/ (accessed on 15 August 2021).

## 3. Results

### 3.1. Baseline Characteristics

Of the 119 patients (87 men, 73.1%; mean age 65.8 ± 14.9 years; body mass index 24.4 ± 15.6 kg/m^2^), 46 (38.7%) survived and 73 (61.3%) did not survive. Overall, 107 patients (89.9%) were smokers or ex-smokers, 20 patients (16.8%) had pre-existing diabetes mellitus (DM), and 39 patients (32.8%) had a diagnosed solid cancer. The mean APACHE II score was 28.7 ± 7.3. As shown in Table 1, significantly more (*p* = 0.03) of the non-survivors than the survivors had a pre-existing solid tumor, with frequencies (percentages) of 28 (38.4%) and 11 (23.9%), respectively. A DM diagnosis was significantly less common among the non-survivors than among the survivors, with frequencies (percentages) of 4 (5.5%) and 16 (34.8%), respectively (*p* < 0.0001).

### 3.2. Clinical Presentation at Time of Isolation and Outcomes

The non-survivors presented with significantly lower absolute lymphocyte counts (ALCs) (612.2 ± 574.5 vs. 799.3 ± 563.1 cells/µL, *p* < 0.001) and significantly higher procalcitonin concentrations (13.2 ± 22.11 vs. 1.8 ± 2.5 ng/mL, *p* = 0.02) than did the survivors. Regarding the ventilator settings, the fraction of inspired oxygen (FiO_2_) was significantly higher for the non-survivor group than for the survivor group (55.3% ± 20.8% vs. 45.3% ± 13.0%, *p* = 0.009), but the tidal volume, positive end-expiratory pressure, and positive inspiratory pressure did not differ significantly between the groups (Table 2). The non-survivors had significantly longer ICU stays (35.8 ± 28.7 vs. 23.2 ± 20.3 days, *p* = 0.01) and ventilation durations (32.2 ± 28.4 vs. 20.8 ± 24.5 days, *p* = 0.03) than the survivors. Both the length of ICU stay (25.0 ± 26.1 vs. 15.5 ± 19.7 days, *p* = 0.04) and the duration of mechanical ventilation (21.4 ± 26.8 vs. 14.4 ± 26.8 days, *p* = 0.03) following isolation were longer in the non-survivors. Furthermore, ARDS was significantly more common among the non-survivors than the survivors, with frequencies (percentages) of 52 (71.2%) and 10 (21.7%), respectively (*p* < 0.0001). AKI was also significantly more common among the non-survivors, with frequencies (percentages) of 58 (79.4%) and 17 (37.0%), respectively (*p* < 0.0001).

The significant factors predictive of mortality during the ICU stay from the univariable Cox regression were entered into the multivariable Cox regression model. As presented in Table 3, these factors were DM diagnosis (odds ratio [OR]: 0.12, 95% confidence interval [CI]: 0.03–0.36, *p* = 0.0003), ALC (OR: 0.99, 95% CI: 0.997–0.999, *p* = 0.007), ARDS diagnosis (OR: 8.91, 95% CI: 3.88–22.1, *p* < 0.0001), AKI diagnosis (OR: 6.6, 95% CI: 2.95–15.46, *p* < 0.0001), APACHE II score (OR: 1.11, 95% CI: 1.05–1.18, *p* = 0.0003), length of ICU stay (OR: 1.02, 95% CI: 1.01–1.04, *p* = 0.02), duration of mechanical ventilation (OR: 1.02, 95% CI: 1.00–1.04, *p* = 0.04), and FiO_2_ setting used for mechanical ventilation (OR: 1.04, 95% CI: 1.001–1.007, *p* = 0.04). The independent mortality-associated factors were an ARDS diagnosis (OR: 4.18, 95% CI: 1.05–17.97, *p* < 0.001), APACHE II score (OR: 1.08, 95% CI: 1.00–1.18, *p* = 0.02), and a DM diagnosis (OR: 0.12, 95% CI: 0.02–0.49, *p* = 0.0009).

The length of hospital stay was plotted against mortality on Kaplan–Meier curves according to the diagnosis of DM, ARDS, and APACHE II score (Figure 3). Lower mortality was noted with significance among patients with DM (*p* = 0.047). Patients diagnosed with ARDS (*p* < 0.0001) or those with high APACHE II scores (*p* = 0.032) had an increased mortality rate.

The ROC curves were analyzed using the multivariable Cox regression model. The AUCs of DM diagnosis, ARDS, and APACHE II score were 0.65, 0.75, and 0.69, respectively. The APACHE II score was 30 at the cutoff point for the ROC curve. While combining the three factors, the AUC for predicting mortality of the patients was 0.82 (Figure 4).

## 4. Discussion

Fewer DM diagnoses, higher ARDS incidence, and higher APACHE II scores were noted among the non-survivors than the survivors. ARDS, APACHE II score, and the absence of a DM diagnosis were established as independent mortality-associated factors. The combination of three predictors showed an increased predictive efficacy for mortality.

Overall, 83.2% of the patients did not have DM, 52.1% developed ARDS, and 55.5% had APACHE II scores of ≥30. Other studies have reported lower mortality rates [8,12,16,22,23]. The slightly greater number of patients with ARDS and the high APACHE II scores in this study may have contributed to the high mortality rate of 61.3% (73/119 patients). Among the non-survivors, 94.5% did not have DM, 71.2% developed ARDS, and 64.4% had APACHE II scores of ≥30. The AUC of ROC curve for APACHE II score against mortality was initially 0.69. The raised AUC of 0.82 after the APACHE II score was combined with the other two factors indicated that the diagnosis of DM or ARDS increased the discriminative ability to predict mortality among these patients.

Our findings suggest that DM protects against mortality in critically ill patients with HSV pneumonitis. Although DM has been associated with poor outcomes in critically ill patients, the negative impact of DM is mainly found in patients after surgery [24,25]. In this large meta-analysis, diabetic patients were found to be more likely to have cardiovascular consequences and wound infections following cardiac surgery. On the other hand, DM has been suggested to be a protective factor [26,27,28,29]. In one prospective study, a lower mortality rate in patients with DM was noted among a population of patients experiencing sepsis [28]. In other investigations, patients with DM developed ARDS at lower rates than patients without DM [27,29]. In the present study, 3 of the 20 patients with DM (15%) developed ARDS, whereas 59.6% of the patients (59/99) without DM developed ARDS. In another study, the short-term mortality among patients with DM remained unchanged, even following ARDS development [29]. The reason DM serves as a protective factor remains unclear. The most common reason provided is that patients with DM can tolerate hyperglycemia to a greater extent than patients without DM; therefore, they experience less harm from the blood sugar level fluctuations that can occur during critical illness [30]. Another reason is that diabetic patients have a higher prevalence of HSV-1 infection [31] and previous infection could protect these patients from severe HSV-1 pneumonia. Further investigations should be performed to confirm this mechanism.

In this study, APACHE II scores were employed as an initial measure of clinical severity. Overall, 55.5% of patients (66/119) had APACHE II scores of ≥30. In a longitudinal cohort study, patients in the ICU with higher APACHE II scores were more likely to test positive for HSV via PCR (using throat swabs or tracheal secretions) and were less likely to survive [7]. In a retrospective study, the mortality rate was higher among patients with APACHE II scores of >15 [4]. However, that study may be unrepresentative of all critically ill patients because it focused on patients with solid cancer, most of whom were immunocompromised and not receiving ventilation. In the present study, high APACHE II scores constituted a significant risk factor for mortality according to the multivariable Cox regression model. Specifically, a higher mortality rate was discovered in patients with APACHE II scores exceeding 30. This suggests that the APACHE II scoring system is a reliable predictor of mortality in patients with HSV pneumonitis, especially in those requiring ICU care.

The coincidence of ARDS diagnosis and HSV detection in the lower respiratory tract among critically ill patients has been researched extensively [5,11,32]. Patients with ARDS and whose tracheobronchial secretions revealed the presence of HSV have been discovered to require ventilator support for longer durations [5] and exhibit higher mortality rates [32]. Other investigations have reported longer durations of ventilator support, venovenous extracorporeal membrane oxygenation (VV-ECMO) support, ICU stay, and hospital stay in patients with severe ARDS requiring VV-ECMO support in the event of HSV reactivation [33,34]. Consistent with the findings of relevant studies, the incidence of ARDS in this study was 52.1% (62/119). Among the patients who did or did not develop ARDS, 83.9% and 36.8% did not survive, respectively. Among the patients with ARDS, 64.5% had APACHE II scores of ≥30, potentially explaining the higher mortality rate in our study than in others [5]. Furthermore, ARDS was an independent mortality-associated factor. The non-survivors in this study had a higher oxygen demand, as determined from the FiO_2_ settings, but no significant differences were detected in the other mechanical ventilation parameters. This finding is similar to that of a study involving 23 intubated patients with HSV pneumonitis [8]. Although the researchers suggested that the pneumonitis did not cause severe pulmonary damage, our study reveals that a greater proportion of non-survivors develop ARDS, suggesting that ARDS and impaired prognosis is tightly bound to HSV pneumonitis or lower airway HSV reactivation [2].

This study has several limitations. First, given its retrospective design, determining whether the positive HSV results were attributable to the reactivation of HSV was challenging, as was establishing whether the HSV infection contributed to the state of the critical illness itself. HSV can reach the lower airway via different routes, including focal spread from other lung parenchyma, aspiration of particles or secretion from infected patients, reactivation of a primary infection contracted during a young age, or transmitted from vagal ganglion via the vagus nerve [35]. However, one study indicated that HSV-1 infection or latency constitutes a marker of critical illness and noted that it was related to high mortality, even in critically ill patients without pathological confirmation of HSV pneumonitis [8]. Second, due to technical limitations at the time of data collection, the viral load of the lower airway secretions as determined via PCR was not recorded. The viral replication amount was also unable to be acquired from the endothelium of the lower airway via biopsy or autopsy. The data were also unable to confirm whether the virus collected in this study was HSV-1 or HSV-2, although both were reported to cause lower airway infection [36]. Third, because the proportion of patients receiving acyclovir was low, we could not evaluate an antiviral treatment response. To verify the predictive efficacy of the proposed factors, large-scale prospective studies are warranted.

## 5. Conclusions

Mortality among ICU patients with HSV pneumonitis can be negatively predicted via a DM diagnosis, and positively predicted by an ARDS diagnosis and high (i.e., ≥30) APACHE II score on ICU admission.

## Figures and Tables

**Figure 1 viruses-14-00205-f001:**
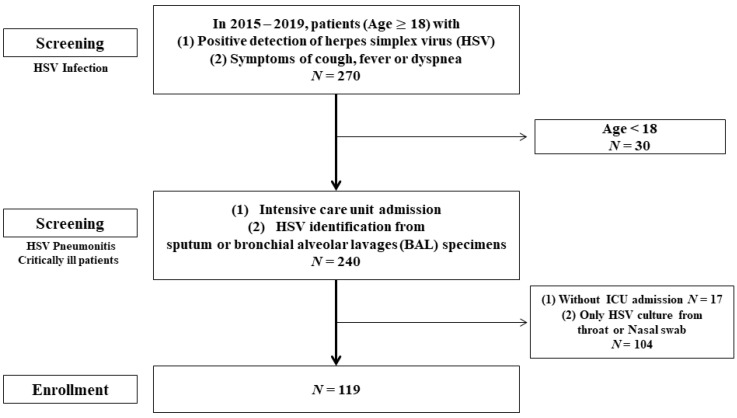
Study flowchart.

**Figure 2 viruses-14-00205-f002:**
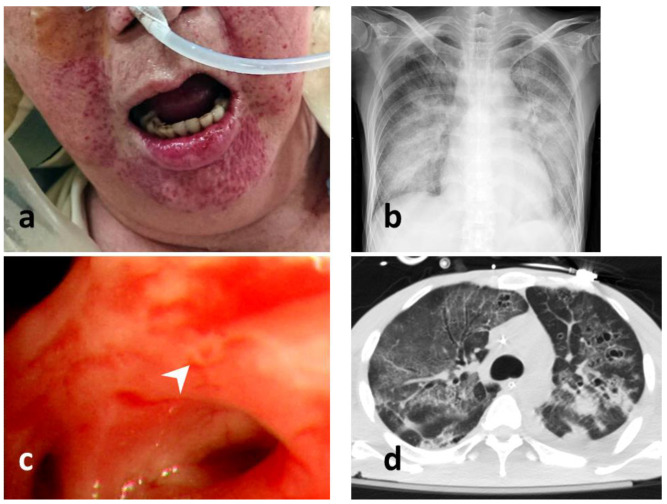
Examinations of HSV infection. (**a**) perioral mucocutaneous lesion of HSV infection; (**b**) a chest radiograph of a patient with HSV pneumonitis; (**c**) bronchoscopic view of HSV pneumonitis with an endobronchial mucosal ulcer (pointed by a white arrow head); (**d**) computed tomography scan of HSV pneumonitis (lung window view).

**Figure 3 viruses-14-00205-f003:**
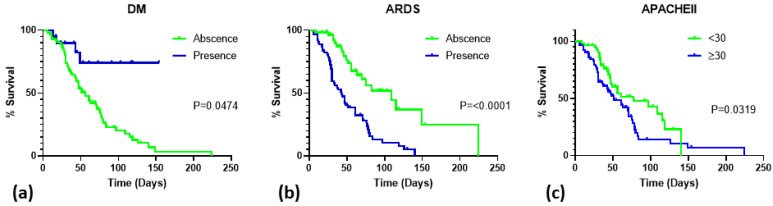
Survival of critically ill patients with HSV pneumonitis. (**a**) survival in relation to DM diagnosis; (**b**) survival in relation to ARDS diagnosis; (**c**) survival in relation to APACHE II score. Abbreviations: DM, diabetes mellitus; ARDS, acute respiratory distress syndrome; APACHE II, Acute Physiology and Chronic Health Evaluation II.

**Figure 4 viruses-14-00205-f004:**
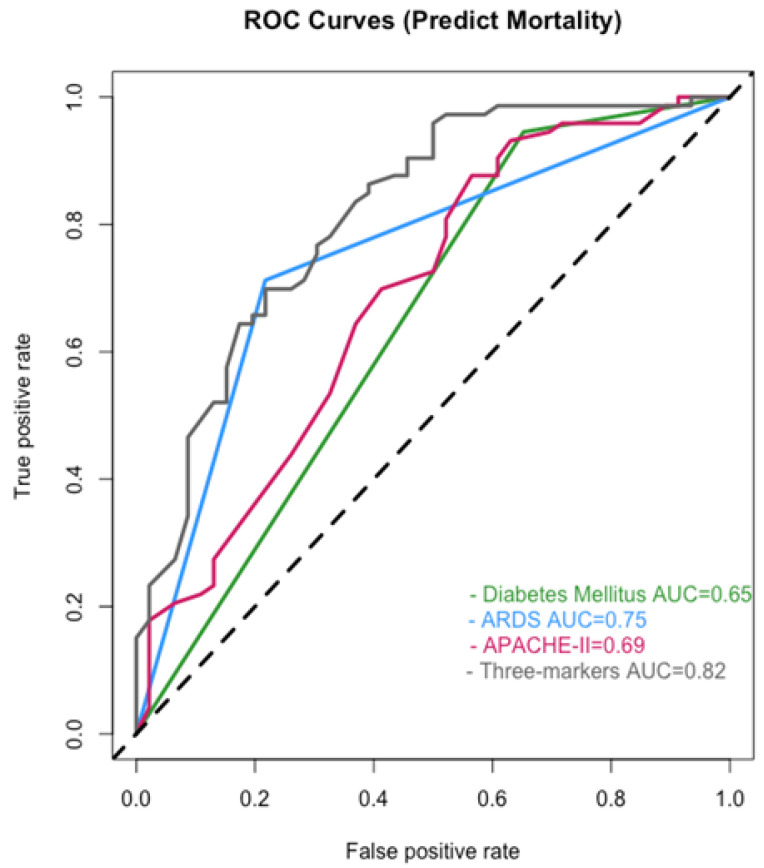
The ROC curves analyzed using the multivariable Cox regression model. Abbreviations: ARDS, acute respiratory distress syndrome; APACHE II, Acute Physiology and Chronic Health Evaluation II; AUC, the area under the curve. The 45-degree black dotted line represents a curve of random classifier.

**Table 1 viruses-14-00205-t001:** Clinical characteristics of all patients, survivors, and non-survivors.

Variable	All Patients	Survivors	Non-Survivors	
	*N* = 119	*N* = 46	*N* = 73	*p*-value
Age, year	65.8 ± 14.9	66.3 ± 12.6	65.4 ± 16.5	0.45
Male, *n* (%)	87 (73.1)	33 (71.7)	54 (74.0)	0.83
Female, *n* (%)	32 (26.9)	13 (28.3)	19 (26.0)	0.83
BMI, kg/m^2^	24.4 ± 15.6	23.6 ± 4.7	25.0 ± 20.3	0.85
Smoker or ex-smokers, *n* (%)	107 (89.9)	43 (93.5)	64 (87.7)	0.24
Underlying diseases				
Receiving steroids or Immunosuppressant agents, *n* (%)	60 (50.4)	19 (41.3)	41 (56.2)	0.13
HIV, *n* (%)	3 (2.5)	0 (0)	3 (4.1)	0.12
Burn, *n* (%)	1 (0.8)	1 (2.2)	0 (0)	0.25
Solid tumors, *n* (%)	39 (32.8)	11 (23.9)	28 (38.4)	0.03
Autoimmune disease, *n* (%)	19 (16.0)	9 (19.6)	10 (13.7)	0.73
Hematologic disease, *n* (%)	18 (15.1)	7 (15.2)	11 (15.1)	0.67
Diabetes mellitus, *n* (%)	20 (16.8)	16 (34.8)	4 (5.5)	<0.0001
Chronic heart disease, *n* (%)	14 (11.8)	5 (10.9)	9 (12.3)	1.00
Chronic lung disease, *n* (%)	3 (2.5)	0 (0)	3 (4.1)	0.28
Chronic liver disease, *n* (%)	9 (7.6)	2 (4.3)	7 (9.6)	0.48
Chronic renal disease, *n* (%)	11 (9.2)	4 (8.7)	7 (9.6)	1.00
Diagnosis of HSV				
HSV alone, *n* (%)	22 (18.5)	12 (26.1)	10 (13.7)	0.09
Combine bacteria, *n* (%)	71 (59.7)	25 (54.3)	46 (63.0)	0.44
Combine fungus, *n* (%)	30 (25.2)	10 (21.7)	20 (27.4)	0.53
Combine virus, *n* (%)	36 (30.3)	9 (19.6)	27 (37.0)	0.06
Combine PJP, *n* (%)	17 (14.3)	4 (8.7)	13 (17.8)	0.19
Combine TB/NTM, *n* (%)	2 (1.7)	0 (0)	2 (2.7)	0.52
Reason for admission				
Respiratory insufficiency, *n* (%)	2 (1.7)	1 (2.2)	1 (1.4)	1.00
Pneumonitis, *n* (%)	80 (67.2)	23 (50)	57 (78.1)	0.002
Sepsis, *n* (%)	32 (26.9)	18 (39.1)	14 (19.2)	0.02
Cardiovascular crisis, *n* (%)	4 (3.4)	3 (6.5)	1 (1.4)	0.30
Shock, *n* (%)	3 (2.5)	3 (6.5)	0 (0)	NA
Renal insufficiency, *n* (%)	0 (0)	0 (0)	0 (0)	NA
Neurologic crisis, *n* (%)	1 (0.8)	1 (2.2)	0 (0)	0.39
APACHE II	28.7 ± 7.3	25.5 ± 7.9	30.7 ± 6.2	<0.0001

Note: Data are presented as means ± standard deviations or as frequencies and percentages. The differences between patients in the survivor and non-survivor groups are considered significant at *p* < 0.05. Abbreviations: BMI, body mass index; HIV, human immunodeficiency virus; HSV, herpes simplex virus; PJP, *pneumocystis jirovecii* pneumonia; TB, *Mycobacterium tuberculosis*; NTM, nontuberculous mycobacteria; APACHE II, Acute Physiology and Chronic Health Evaluation II; NA, not applicable due to lack of samples on either group.

**Table 2 viruses-14-00205-t002:** Clinical signs and ventilator settings at the time of HSV pneumonitis diagnosis.

Variable	Total Patients	Survivors	Non-Survivors	
	*N* = 119	*N* = 46	*N* = 73	*p*-value
Chest radiograph findings				
GGO, *n* (%)	15 (12.6)	7 (15.2)	8 (11.0)	0.57
Interstitial pattern, *n* (%)	33 (27.7)	10 (21.7)	23 (31.5)	0.30
Consolidation, *n* (%)	68 (57.1)	29 (63)	39 (53.4)	0.34
Laboratory data				
ALC, cells/μL	684.5 ± 575.0	799.3 ± 563.1	612.2 ± 574.5	<0.0001
CRP, mg/dl	119.9 ± 107.4	110.1 ± 126.4	78.2 ± 122.7	0.43
Procalcitonin	9.0 ± 18.4	1.8 ± 2.5	13.2 ± 22.1	0.02
HSV specific presentation				
Oral-labial lesion, *n* (%)	69 (58.0)	25 (54.3)	44 (60.3)	0.57
Macroscopic bronchial lesions, *n* (%)	66 (75.9)	25 (73.5)	41 (77.4)	0.79
Positive of HSV IgG serology, *n* (%)	13 (10.9)	3 (6.5)	10 (13.7)	0.10
Treatment (Yes), *n* (%)	41(23.8)	15 (20)	26 (26.8)	0.37
Treatment duration	10.6 ± 5.7	11.0 ± 7.1	10.4 ± 4.9	0.78
Ventilator setting				
FiO^2^ %	51.5 ± 18.8	45.3 ± 13.0	55.3 ± 20.8	0.009
A-a gradient, mmHg	210.2 ± 121.7	180.3 ± 102.2	228.5 ± 129.6	0.07
Hospital LOS, days	53.3 ± 35.9	56.7 ± 32.6	51.1 ± 38.0	0.24
Total ICU LOS, days	31.0 ± 26.4	23.2 ± 20.3	35.8 ± 28.7	0.01
ICU LOS prior to isolation, days	9.2 ± 12.3	7.3 ± 5.6	10.5 ± 4.9	0.26
ICU LOS after isolation, days	21.4 ± 24.3	15.5 ± 19.7	25.0 ± 26.1	0.04
Ventilation duration	27.7 ± 27.4	20.8 ± 24.5	32.2 ± 28.4	0.03
Ventilation duration after isolation, days	18.6 ± 26.7	14.4 ± 26.8	21.4 ± 26.8	0.03
Organ failure				
ARDS, *n* (%)	62 (52.1)	10 (21.7)	52 (71.2)	<0.0001
AKI, *n* (%)	75 (63.0)	17 (37.0)	58 (79.5)	<0.0001
Reason for mortality				
Respiratory failure, *n* (%)	22 (18.5)		22 (30.1)	
Septic shock	25 (21.0)		25 (34.2)	
Solid tumors	5 (4.2)		5 (6.8)	
MOF	21 (17.6)		21 (28.8)	

Note: Data are presented as means ± standard deviations or as frequencies (percentages). The differences between the survivor and non-survivor groups are significant at *p* < 0.05. Abbreviations: GGO, ground-glass opacity; ALC, absolute lymphocyte count; CRP, C-reactive protein; ICU, intensive care unit; PEEP, positive end-expiratory pressure; PIP, positive inspiratory pressure; FiO_2_, fraction of inspired oxygen; LOS, length of stay; ARDS, acute respiratory distress syndrome; AKI, acute kidney injury; MOF, multiple organ failure.

**Table 3 viruses-14-00205-t003:** Univariable and multivariable logistic regression models for mortality.

Variable	Univariable Analysis			Multivariable Analysis		
	OR	95%CI	*p*-value	OR	95%CI	*p*-value
Age, per year	1	0.98–1.02	0.99			
Gender	1.44	0.69–3.03	0.33			
BMI	0.99	0.92–1.07	0.83			
Smoker	1.55	0.75–3.24	0.24			
Underlying diseases						
Receiving steroids/immunosuppressive agents	1.68	0.86–3.37	0.13			
HIV	1.64	0.15–35.88	0.69			
Burn	3.71	2.42–5.05	0.99			
Solid tumors	1.78	0.84–3.90	0.14			
Autoimmune disease	0.79	0.24–2.68	0.71			
Hematologic disease	1.25	0.43–3.93	0.67			
Diabetes mellitus	0.12	0.03–0.36	0.0003	0.12	0.02–0.49	0.0009
Chronic organ disease	1.78	0.84–3.90	0.14			
Laboratory data						
ALC	0.99	0.997–0.999	0.007	0.99	0.99–1.01	0.76
CRP	1.00	0.99–1.01	0.1			
Procalcitonin	1.11	1.02–1.36	0.11			
Clinical presentation						
Oral labial lesions	1.21	0.61–2.39	0.59			
Macroscopic bronchial lesions	1.50	0.59–3.84	0.39			
Hospital LOS	0.99	0.99–1.00	0.42			
ICU LOS	1.02	1.01–1.04	0.02	1.01	0.97–1.05	0.81
Ventilation duration	1.02	1.00–1.04	0.04	1.01	0.97–1.05	0.73
Ventilator setting						
PEEP	1.03	0.84–1.27	0.78			
Driving pressure	1.06	0.99–1.15	0.09			
PIP	1.07	0.09–1.15	0.07			
Tidal volume	0.99	0.99–1.00	0.41			
FiO^2^	1.04	1.01–1.07	0.008	1.03	0.99–1.08	0.10
A-a gradient	1.00	1.001–1.007	0.04	0.99	0.99–1.00	0.47
Treatment (Yes)	0.96	0.42–2.21	0.91			
Organ failure						
ARDS	8.91	3.88–22.1	<0.0001	4.18	1.05–17.97	<0.0001
AKI	6.6	2.95–15.46	<0.0001	2.70	0.75–9.72	0.11
ALI	5.54	0.97–104.56	0.11			
APACHE II scores	1.11	1.05–1.18	0.0003	1.08	1.00–1.18	0.02

Abbreviations: BMI, body mass index; HIV, Human immunodeficiency virus; HSV, herpes simplex virus; CRP, C-reactive protein; ALC, absolute lymphocyte count; ICU, intensive care units; PEEP, positive end expiratory pressure; PIP, positive inspiratory pressure; FiO_2_, fraction of inspired oxygen; LOS, length of stay; ARDS, acute respiratory distress syndrome; AKI, acute kidney injury; ALI, acute liver failure; APACHE II, Acute Physiology and Chronic Health Evaluation II.

## Data Availability

The datasets analyzed during the current study are available from the corresponding author upon reasonable request.

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
