# Peer review of "Outcomes of Herpes Simplex Virus Pneumonitis in Critically Ill Patients"

_viruses, 2022, doi:10.3390/v14020205_

Round 1
Reviewer 1 Report
In this retrospective study, Chang et al looked at several comorbidities and pre-existing conditions as possible contributors/predictors of mortality during herpes simplex virus (HSV) pneumonitis in hospitalized patients. The study included 119 patients in the Chang Gung Memorial Hospital in Taipei, Taiwan. The selection criteria included positive HSV sample form the lower respiratory tract, ICU stay, and age >18 years. An extensive list of factors and underlying conditions such as chronic disease, immunosuppressant therapy, tumors, ventilator settings and laboratory results were considered as possible factors contributing to mortality.
The authors found that mortality was associated with lower absolute lymphocyte counts, longer ICU stays, acute kidney injury and ARDS diagnoses, higher APACHE II scores and longer need for ventilation with higher oxgen demands. Analysis of ROC curves using multivariable Cox regression model, the authors conclude that DM has a protective role against death during HSV pneumonitis. They further conclude that ARDS diagnosis and APACHEII score ≥30, and absence of DM were independent predictors of increased mortality during HSV pneumonitis.
The manuscript is well written and several criteria were considered.
Few minor issues could be addressed before publication
- Line 64: Name of the hospital is mentioned, but the city and country should also be mentioned.
- Introduction could be expanded to discuss the prevalence of HSV pneumonitis in critically ill patients
- How is HSV pneumonitis contracted- is it typically a re-activation event from a latent reservoir such as trigeminal ganglion? This could be mentioned briefly in the introduction or discussion sections.
- Table 1 could mention the numbers and percentages for women, instead of having the reader figure them out based on the numbers for men.
- The finding that DM has a protective role was interesting. The authors mention one possible explanation, but this section in the discussion could be expanded to discuss potential mechanisms for both possibilities (DM as a contributing factor to death vs protective role). What reasons did the other studies discuss?
Author Response
Thank you for sending back our manuscript together with the comments of the Reviewers and your letter. We thank you for their valuable comments, and we have responded to all the comments made point-by-point. We have made appropriate changes to the comments made and corrected the mistakes pointing out by the Reviewers. We enclose a revised manuscript which incorporates these changes by the red color.
Reviewer 1:
- Line 64: Name of the hospital is mentioned, but the city and country should also be mentioned.
Response: The city and country of the hospital was added to the manuscript.
- Introduction could be expanded to discuss the prevalence of HSV pneumonitis in critically ill patients
Response: Thank you for the important suggestion for the description we have missed. Two prospective studies were included in the introduction describing the prevalence of HSV detection in ICU patients.
- How is HSV pneumonitis contracted- is it typically a re-activation event from a latent reservoir such as trigeminal ganglion? This could be mentioned briefly in the introduction or discussion sections.
Response: The HSV pneumonitis or HSV detected in lower airway could be due to the aspiration of infected particles or reactivation of latency from vagal ganglion. The description was added in the discussion. Thank you for your comment.
- Table 1 could mention the numbers and percentages for women, instead of having the reader figure them out based on the numbers for men.
Response: We have adjusted the issue in Table 1. We will make sure to present data of both biological gender in all tables of our future study. Thank you very much.
- The finding that DM has a protective role was interesting. The authors mention one possible explanation, but this section in the discussion could be expanded to discuss potential mechanisms for both possibilities (DM as a contributing factor to death vs protective role). What reasons did the other studies discuss?
Response: Thank you for the precise commentary. We updated the mechanism of DM as a contributing factor to death in the manuscript as following: “Although DM has been associated with poor outcomes in critically ill patients, the negative impact of DM is mainly on patients after surgery [24,25]. In the large meta-analysis, diabetic patients are more likely to have consequences of cardiovascular consequences and wound infections after cardiac surgery.” We also tried to explain this condition by assuming that previous infection could protect these patients from severe HSV pneumonitis. However, there was little evidence showing that development of HSV pneumonitis could be protected by previous infection with the same virus. The IgG against HSV was also not strongly suggested as a marker for immunity to HSV infection. A prospective study might be required to accomplish firm explanation of this phenomenon.
Reference:
- Sun Y, Pei W, Wu Y, Yang Y. An association of herpes simplex virus type 1 infection with type 2 diabetes. Diabetes care 2005; 28(2): 435-6

Reviewer 2 Report
Authors in this study designed a very well comprehending report about critically ill HSV patients developing HSV pneumonitis. The sputum samples of infected patients were collected for HSV detection. Standard protocols have been followed to perform experiments. The study is well designed, interesting and provides valuable information. However, following issues needs to be addressed.
- Please provide the clear relationship between HSV infection and Diabetes mellitus.
- Did the authors measure virus replication in the infected tissues?
- Authors need to state the relevance of this study, perhaps needs to mention the epidemiological status of HSV pneumonitis in humans.
- Line 211-212, please provide possible hypothesis for the mechanism underlying DM protection against HSV infection.
- Line 255, please provide the reference.
- Finally, Authors need to clarify the type of HSV infection they are studying.
Author Response
Reply to the Editor's Comments
Thank you for sending back our manuscript together with the comments of the Reviewers and your letter. We thank you for their valuable comments, and we have responded to all the comments made point-by-point. We have made appropriate changes to the comments made and corrected the mistakes pointing out by the Reviewers. We enclose a revised manuscript which incorporates these changes by the red color.
- Please provide the clear relationship between HSV infection and Diabetes mellitus.
- Line 211-212, please provide possible hypothesis for the mechanism underlying DM protection against HSV infection.
Response: Thank you for the precise commentary. We updated the mechanism of DM as a contributing factor to death in the manuscript as following: “Although DM has been associated with poor outcomes in critically ill patients, the negative impact of DM is mainly on patients after surgery [24,25]. In the large meta-analysis, diabetic patients are more likely to have consequences of cardiovascular consequences and wound infections after cardiac surgery.” Patients diagnosed with diabetes were reported to be vulnerable to primary infection of HSV, and with higher chance to acquire IgG against HSV-1[1]. However, whether this clinical condition provides protection was not confirmed and the supporting data was yet to be collected.
Reference:
- Sun Y, Pei W, Wu Y, Yang Y. An association of herpes simplex virus type 1 infection with type 2 diabetes. Diabetes care 2005; 28(2): 435-6
- Did the authors measure virus replication in the infected tissues?
Response: Due to the limitation of retrospective study, the biopsy or autopsy was not able be arranged. We have added related new description in the discussion. Thank you for the kind reminder.
- Authors need to state the relevance of this study, perhaps needs to mention the epidemiological status of HSV pneumonitis in humans.
Response: Thank you for the suggestion. We have added two prospective studies for the discussion of prevalence of HSV detection in ICU patients. This can be found in the introduction.
- Line 255, please provide the reference.
Response: Thank you for this important suggestion. We have modified our way of description and updated the reference.
- Finally, Authors need to clarify the type of HSV infection they are studying.
Response: Due to being a retrospective study, our investigation was unable to acquire further data of each positive result of HSV culture or PCR being HSV-1 or HSV-2. Although in CGMH, the culture and PCR finding of HSV were restricted to HSV-1 and HSV-2. This limitation was added to the discussion.
Reviewer 3 Report
The authors present an interesting study which reports the outcomes of patients will human alphaherpesvirus 1 (HHV-1) associated pneumonitis and has characterised associations with multiple clinical parameters with mortality. Interestingly, patients with diabetes mellitus were more likely to survive, while those with adult respiratory distress syndrome (ARDS) were not.
Of the three factors associated with mortality in the multivariate analyses:
Diabetes Mellitus - OR 0.12 (95%CI 0.02-0.51, p = 0.0009)
ARDS - OR 3.18 (95%CI 0.86-12.19, p <0.0001)
APACHE II scores – OR 1.08 (95%CI 1.00-1.18, p = 0.02)
The 95%CI suggests that most of the variation in the APACHE II scores is explained in the modelling. In contrast, the 95%CIs for diabetes mellitus and ARDS are wide, suggesting there are potentially other factors not included in the models that are affecting patient risk. The authors should discuss this in the relevant sections of the discussion. Similarly, I am perplexed by the 95%CI of ARDS spanning “1”, i.e. “0.86-12.19” with such robust statistical support (p <0.0001). This suggests an HSV pneumonitis patient with ARDS can be 20 times less likely to die and also be 12 times more likely to die at the extremes of the model. I would ask the authors to check their modelling -any outcome of statistical modelling must make biological sense. In any case, the author should discuss the potential implications of wide 95%CI.
Virus nomenclature: the authors refer to the virus of interest as Herpes Simplex Virus (HSV) which is of course the widely used name for the virus of interest. However, the correct taxonomic reference for the virus is human alphaherpesvirus 1 (HHV-1). Given the clinical nature of this manuscript, I would not suggest using strict taxonomic names. I would suggest the authors refer to this name in the introductory text as a compromise. One point which is unclear is there are two herpes simplex viruses, HSV-1 and HSV-2. While HSV-1 and HSV-2 are generally associated with oral/respiratory and sexually transmitted diseases, respectively, these associations are not absolute. Which raises the question of whether HVS-2 has ever been associated with a case of pneumonitis? If it has, were the diagnostic tests used in the current study able to differentiate between the two viruses? What I am suggesting is if the authors “know” a specific virus is associated with HSV pneumonitis the manuscript should clearly state this.
Lines 17 to 30 – As the abstract is considered a separate document from the main text all abbreviations must be explained in full and kept to a minimum.
Line 34 suggest revision “is associated with various infections”
Line 124 suggest revision “had a diagnosed solid cancer”
Line 256 suggest revision “its retrospective design”
Line 259 Please review the use of “colonization” – this is not a term used in virology, particularly differentiated to “infection”. Are the authors referring to “latency”?
Line 34 suggest revision “is associated with various infections”
Author Response
Reply to the Editor's Comments
Thank you for sending back our manuscript together with the comments of the Reviewers and your letter. We thank you for their valuable comments, and we have responded to all the comments made point-by-point. We have made appropriate changes to the comments made and corrected the mistakes pointing out by the Reviewers. We enclose a revised manuscript which incorporates these changes by the red color.
- The authors present an interesting study which reports the outcomes of patients will human alpha herpesvirus 1 (HHV-1) associated pneumonitis and has characterized associations with multiple clinical parameters with mortality. Interestingly, patients with diabetes mellitus were more likely to survive, while those with adult respiratory distress syndrome (ARDS) were not.
Of the three factors associated with mortality in the multivariate analyses:
Diabetes Mellitus - OR 0.12 (95%CI 0.02-0.51, p = 0.0009)
ARDS - OR 3.18 (95%CI 0.86-12.19, p <0.0001)
APACHE II scores – OR 1.08 (95%CI 1.00-1.18, p = 0.02)
The 95%CI suggests that most of the variation in the APACHE II scores is explained in the modelling. In contrast, the 95%CIs for diabetes mellitus and ARDS are wide, suggesting there are potentially other factors not included in the models that are affecting patient risk. The authors should discuss this in the relevant sections of the discussion. Similarly, I am perplexed by the 95%CI of ARDS spanning “1”, i.e. “0.86-12.19” with such robust statistical support (p <0.0001). This suggests an HSV pneumonitis patient with ARDS can be 20 times less likely to die and also be 12 times more likely to die at the extremes of the model. I would ask the authors to check their modelling -any outcome of statistical modelling must make biological sense. In any case, the author should discuss the potential implications of wide 95%CI.
Response: Thank you for the crucial comment on statistics. We have rechecked our table and the outcome of R studio codes. The development of ARDS was less likely to contribute to good prognosis. An error during the multivariate analysis was noted and were corrected as showed in the table.
- Virus nomenclature: the authors refer to the virus of interest as Herpes Simplex Virus (HSV) which is of course the widely used name for the virus of interest. However, the correct taxonomic reference for the virus is human alphaherpesvirus 1 (HHV-1). Given the clinical nature of this manuscript, I would not suggest using strict taxonomic names. I would suggest the authors refer to this name in the introductory text as a compromise. One point which is unclear is there are two herpes simplex viruses, HSV-1 and HSV-2. While HSV-1 and HSV-2 are generally associated with oral/respiratory and sexually transmitted diseases, respectively, these associations are not absolute. Which raises the question of whether HVS-2 has ever been associated with a case of pneumonitis? If it has, were the diagnostic tests used in the current study able to differentiate between the two viruses? What I am suggesting is if the authors “know” a specific virus is associated with HSV pneumonitis the manuscript should clearly state this.
Response: Thank you for suggesting this important issue. As mentioned in the sixth question of Reviewer 2, due to the retrospective design of our study, the investigation was unable to include further data of each positive result of HSV culture or PCR being HSV-1 or HSV-2. In our hospital, CGMH, the culture and PCR finding of HSV were restricted to HSV-1 and HSV-2. The condition might also be due to the patient data was collected from those who were admitted from 2015-2019. The devices and examination kit might be unable to provide this information back then. Further reports and details could be available in the future if the laboratory were upgraded. This limitation was added to the discussion.
- Lines 17 to 30 – As the abstract is considered a separate document from the main text all abbreviations must be explained in full and kept to a minimum.
Response: Thank you for this kind reminder. The incomplete and inconsistent abbreviations in the abstract were revised.
- Line 34 suggest revision “is associated with various infections
Response: Thanks for the suggestion. We updated it in the manuscript.
- Line 124 suggest revision “had a diagnosed solid cancer”
Response: Thanks for the suggestion. We updated it in the manuscript.
- Line 256 suggest revision “its retrospective design”
Response: Thanks for the suggestion. We updated it in the manuscript.
- Line 259 Please review the use of “colonization” – this is not a term used in virology, particularly differentiated to “infection”. Are the authors referring to “latency”?
Response: Thanks for the suggestion. We updated it in the manuscript.
- Line 34 suggest revision “is associated with various infections”
Response: Thanks for the suggestion. We updated it in the manuscript.